# Tumor Treating Fields and Combination Therapy in Management of Brain Oncology

**DOI:** 10.3390/cancers17071211

**Published:** 2025-04-02

**Authors:** Ruisi Nicole Liu, James H. Huang, Xiaoming Qi, Yizhong Pan, Erxi Wu, Damir Nizamutdinov

**Affiliations:** 1Department of Neurosurgery and Neuroscience Institute, Baylor Scott & White Health, Temple, TX 76508, USA; 2Department of Neurology, Baylor Scott & White Health, Temple, TX 76508, USA; 3Department of Neurosurgery, First Affiliated Hospital of Soochow University, Suzhou 215005, China; 4Department of Neurosurgery, Baylor College of Medicine, Temple, TX 76508, USA

**Keywords:** tumor treating fields (TTF), glioblastoma, tumor combination therapy, brain cancer, tumor survival

## Abstract

This comprehensive review outlines the current knowledge about tumor treating fields in the management of brain tumors, including glioblastoma. This technology is utilized in various applications and combination with other established, standard, and/or novel therapeutic modalities. This communication guides readers seeking critical points related to treatment using tumor treating field technology in combination with other approaches, as well as their biological mechanisms, efficacy, side effects, and challenges and the most common limitations that professionals face in clinical settings when managing patients with devastating diseases. This review covers essential topics related to this transformative therapeutic modality and discusses future directions.

## 1. Introduction

Malignant gliomas, originating from glial cells, are the most common and aggressive tumor types among primary central nervous system cancers, with high morbidity and mortality [1,2]. The most common types of malignant gliomas are glioblastomas (GBMs), anaplastic astrocytomas, oligodendrogliomas, and oligoastrocytomas [3,4]. According to the Central Brain Tumor Registry of the United States (CBTRUS), 14.2% of all tumors and 50.1% of all malignant tumors are GBM [2]. While GBM can develop at any age, the likelihood of occurrence increases with age and it is more often seen in the male sex than the female sex [2]. The median survival of patients with GBM remains around 14 months, with a 5-year survival rate of 6.9% [2,5]. The high invasiveness of GBM essentially leads to tumor recurrence, treatment resistance, and low survival rates [6]. With the currently available treatments, for patients who underwent primary debulking surgery followed by chemo-radiotherapy and 6-month adjuvant chemotherapy with temozolomide (TMZ), the 1-year survival rate could reach as high as 30% and there was a 2-year survival rate of 6.7%. Even so, patients’ overall survival (OS) and quality of life remained poor [5].

Multiple treatment methods can be used to treat GBM. The most common treatment types include surgery, radiotherapy, chemotherapy, targeted therapy, and immunotherapy [7,8]. Among all, TMZ is the regular approach for chemotherapy, but it is limited by drug resistance [8]. Gross total resection improves patients’ OS and progression-free survival compared to subtotal resection, but the quality of supporting evidence is not high [9]. While standard treatments generally improve median OS for newly diagnosed GBM patients, they do not appear to prolong survival for recurrent GBM patients [1]. Data also suggest that these treatment methods improve quality of life and increase OS. They tend to lead to severe side effects with unbearable consequences [7,8]. Elderly patients especially may not tolerate full-course chemotherapy and surgical resection, urging the need for improvements in the cancer treatment field [7,10].

Recent preclinical studies have introduced several innovative and alternative therapies for GBM, aiming to enhance treatment efficacy and patient outcomes. Notable advancements include the following: (1) Drug repurposing strategies—repositioning existing medications for new therapeutic purposes—offer a promising approach. Studies have explored the use of the Comprehensive Undermining of Survival Paths (CUSP9) protocol, which combines temozolomide with nine repurposed drugs: aprepitant, captopril, auranofin, celecoxib, itraconazole, disulfiram, minocycline, quetiapine, and sertraline. This combination targets GBM growth pathways, showing potential in preclinical models [11,12]. (2) Sonodynamic therapy utilizes ultrasound with a photosensitizer to target cancer cells. Preclinical studies have demonstrated that low-intensity pulsed ultrasound, when combined with systemically administered drugs, increases brain tissue drug concentrations and prolongs the survival of GBM models [13,14,15]. (3) Tumor treating fields (TTFields) involves delivering low-intensity, intermediate-frequency electric fields to disrupt cancer cell division. Preclinical research suggests that TTFields may also be effective in pediatric GBM, with studies indicating its potential efficacy and safety in treating children with high-grade gliomas [16,17]. (4) Oncolytic virus therapy employs modified viruses to infect and selectively destroy tumor cells. Researchers have engineered poliovirus derivatives, such as PVSRIPO, to target GBM cells specifically. Early-phase clinical trials have shown promise, with some patients experiencing prolonged survival after treatment [18,19]. (5) Cell therapy innovations are advancements in cell therapies, including chimeric antigen receptor (CAR) T-cell and CAR natural killer (NK)-cell therapies, involve re-engineering a patient’s immune cells to recognize and attack cancer cells specifically [20,21]. These developments represent the dynamic nature of challenging GBM research and show potential in preclinical studies and offer hopes to improve patient outcomes.

TTFields are a novel, noninvasive antimitotic anti-cancer treatment method using low-intensity (1–3 v/m), intermediate-frequency (~100–500 kHz), and alternating electric fields to target cancer cells [22]. The FDA has recently approved it for treating GBM and malignant pleural mesothelioma (MPM) [23]. In 2004, Kirson et al. presented the possibility of TTFields in repressing tumor growth in vitro (cell culture) and in vivo (mouse models) [24]. Upon applying TTFields, cancer cells exhibited the impaired movement of mitotic spindles during metaphase and incomplete cytokinesis, resulting in cell destruction and apoptosis [24]. At the end of the treatment, the treated tumor size was, on average, only 47% of the control tumor size (n = 78 mice; *p* < 0.001) [24].

Electric fields are found to have a significant effect on cellular activities due to charged cellular structures that are subject to electrical forces, providing clinical advantages in the medical field [7]. Bioelectric circuitry provides resting membrane potential and an endogenous electric field in all cells and tissues. While normal electrically active cells utilize the endogenous electric field to communicate and function properly, cancer cells possess malfunctioning bioelectric circuitry, leading to abnormal cell proliferation and growth [25]. Research evidence suggests that voltage imbalance within cells causes changes in tumor invasion in vitro, tumor metastasis and growth in vivo, and changes in gene expression associated with cell adhesion, which is directly regulated by cellular potassium ion channels [26]. On the other hand, an exogenous electric field stimulates daily biological processes within cells, such as muscle contraction and nervous system transduction [8].

TTFields have become a potential anti-tumor treatment method due to their selective antimitotic effects on dividing cells only [1,24]. The mechanism by which TTFields affect cancer cells primarily involves the inhibition of cell mitosis, which triggers cell self-destruction and apoptosis, activates the anti-tumor immune response, and increases cell membrane permeability (Figure 1).

The figure schematically represents how and in what stages of cell development TTFields affect cancer cell biology, leading to cancer cell death.

## 2. Inhibition of Cell Mitosis

When cells are in the S phase of interphase, cancer cells’ exposure to TTFields reduces the expression of specific DNA repair, replication fork, chromosome maintenance, and cell cycle regulatory genes, thereby decreasing the replication fork speed and increasing DNA replication stress [27,28]. Moreover, applying TTFields increases the formation of R-loops, a DNA-RNA hybrid structure that, when defective, contributes to multiple human diseases [27]. R-loops formation also indicates increased DNA replication stress when cells are in the S phase [27]. When synergistically combined with radiation therapy, TTFields have also been proven to effectively enhance the level of DNA damage during interphase by inhibiting DNA double-strand break repair after radiation, possibly by blocking the homologous recombination repair pathway [29]. Besides interfering with DNA replication, evidence suggests that TTFields exert endoplasmic reticulum and genotoxic stress on proliferating cells [30].

As mentioned above, all cells and tissues possess a bioelectric circuitry that regulates normal cell activities. Especially during mitosis, the rapidly dividing cells contain highly polar and spatially oriented microtubules that respond sensitively to electric fields [24]. During metaphase, as chromosomes line up along the metaphase plate, tubulins organize the mitotic spindle. They must end up in specific arrangements within the cell to form microtubules, marking an essential checkpoint in mitosis [8,28,31]. With the interference of TTFields, the spindles are forced to align with the electric field, disturbing the correct microtubule alignment and tubulin polymerization. With the interruption of appropriate microtubule formation, mitosis is arrested before metaphase is completed [24]. Evidence also suggests that TTFields restrain cell migration and invasion by inducing a more adhesive cell phenotype [1]. As microtubules are forced to align with TTFields, cancer cell migration is hindered upon the activation of GEF-H1/RhoA/ROCK signaling pathways, resulting in the induction of focal adhesion formation and peripheral actin bundling [32]. This characteristic of TTFields potentially reduces the probability of cancer recurrence and metastasis by increasing the size and peripheral distribution of focal adhesions [33].

When transitioning from metaphase to anaphase, cells exposed to TTFields display abnormal membrane blebbing [34]. The mitotic septin complex is a highly polar molecule with a strong dipole moment and plays an essential role in cytokinetic cleavage furrow placement and contraction during cytokinesis [34]. TTFields inhibit the correct localization of the septin complex to the anaphase spindle midline and cytokinetic furrow, resulting in an abnormal nuclear architecture and cellular stress [34]. When TTFields are applied, the dividing cells result in a non-uniform field distribution with all polar organelles being pushed toward the center of the cells at the cytokinetic cleavage furrow, ultimately leading to mitotic catastrophe [8]. Once cancer cells signal abnormal mitotic events after being exposed to TTFields, the autophagosome marker LC-II/LC-I shows an increase in expression, a signal of cell autophagy [8].

## 3. Anti-Tumor Immune Response and Increasing Cell Membrane Permeability

TTFields also activate anti-tumor immune response in cancer cells [35]. When exposed to TTFields, GBM cells exhibit large cytosolic micronuclei clusters [36]. TTFields could potentially serve as a dual and local recruiter/activator of both cGAS/STING and AIM2caspase I inflammasomes, two important pharmaceutical agonists in cancer immunotherapy, through the disruption of the focal nuclear envelope during the S and G2 phases to generate an antitumor immune response against GBM tumors [28,36]. For brain tumors specifically, TTFields have also shown promising evidence of increasing the permeability of the blood–brain barrier (BBB). This barrier often limits the CNS’s delivery of pharmaceuticals by disrupting tight junction proteins [36,37,38]. TTFields also disrupt cancer cell membranes, potentially increasing cell permeability via a combination of the bio-electrorheological and electroporation models [39]. The disruption of cell membranes by TTFields is tumor-specific and does not appear in normal cells [8]. Evidence also suggests that TTFields impair angiogenesis, leading to the repression of tumor growth and development [40]. This is induced by one alpha-regulated pathway through VEGF- and hypoxia-induced factors [28,40]. Another critical point is that TTFields are at their maximum efficiency when the direction of the electric field is parallel to the axis of cell division and minimum efficiency when the two directions are vertical, suggesting that during clinical applications, the placement of the electric field should allow different directions of the electric fields on patients’ scalp to maximize the efficiency of the therapy method [24,41].

## 4. Pre-Clinical Experiments with TTFields

TTFields with a low intensity (<2 V/m) and intermediate frequency (100–300 kHz) were first tested in vitro with cell cultures and in vivo on mice with malignant melanoma and rats with gliomas. Results showed that alternating electric fields effectively slowed down in vitro tumor replications, but the effects of TTFields in one direction were almost negligible. However, data revealed a 42.6% reduction in tumor size for two-directional and a 53.4% reduction in three-directional TTFields, with effects persisting for at least 72 h after treatments [24,41]. No significant adverse effects were observed in the experimental animals. Rabbits with melanoma metastases in the lungs were also tested with TTFields in vivo. Results proved an increase in the median survival of the rabbits and that TTFields might also be able to prevent metastatic spread from primary tumors [42]. Other in vitro pre-clinical experiments demonstrated that the treatment of pancreatic cancer cells with TTFields effectively decreased cancer cell count, increased cell volume, and reduced clonogenicity, suggesting that alternating electric fields disrupted the formation of mitotic spindles, eventually leading to errors in chromosome segregation and cytokinesis [43].

Due to rapidly alternating electric fields, TTFileds have no effects on terminally differentiated cells (i.e., normal adult brain cells), rapidly proliferating cells in regions other than the treated areas, or nerve and muscle cells [1]. Studies on brain tumors have suggested that TTFields may lead to significant changes only at the brain level, including, but not limited to, an increase in BBB permeability, variations in cerebral blood flow, and changes in neurotransmitter concentrations [44]. The optimal intensity and frequency of TTFields depend on the type of cancer cells and can vary significantly [8]. The tumor-inhibitory effects of TTField treatments are also proportional to electric field intensity in a given range [41]. Evidence suggests that the optimal frequency of the electric field is inversely proportional to the size, shape, and cell line of cancer cells [24,41,45]. The electric intensity required to reach target tumor tissues depends strongly on the position of tumors, conductivity, and the impedance of the surrounding tissues [46]. However, a minimal threshold of 1 V/cm has been demonstrated across tumor types [1], and the optimal electric field frequency for ovarian and glioma cells is 200 kHz [47]. Moreover, data suggest that compared to continuous treatment with TTFields at a stable frequency of 200 kHz, changing the frequency from 200 to 150 kHz profoundly increased the inhibitory effect of TTFields (37% ± 10% vs. 57% ± 16% decrease in cell count, respectively; *p* = 0.043) [48]. TTFields have also been proven to induce mutations in tumor cells [48].

## 5. Clinical Experiments

In 2007, Kirson E.D. et al. attempted to treat ten patients with TTFields, seven males and three females. The median OS was 62.2 weeks, and the median time to disease progression was 26.1 weeks, more than double the median of historical control patients with chemotherapy. A total of 90% of patients reported mild to moderate adverse effects on the skin [22]. Meanwhile, a Phase III trial, EF-11, was carried out from September 2006 to May 2009, involving 237 randomized patients [45,49]. Among all, 120 randomly selected patients were treated with TTFields alone, while the remaining 117 control patients received the physician’s best choice of chemotherapy. Patients had to be at least 18 years old to participate in the trial, with World Health Organization grade IV GBM in the supratentorial compartment. They were also required to undergo radiotherapy before treatment plans. Patients with medical device implants, such as pacemakers, were excluded from the study [45,49]. The duration of the TTField treatments consisted of a 4-week cycle with the continuous usage of the device. After each cycle, patients were given a 2–3-day break before the next cycle. Two breaks, each lasting 1 h, were also allowed each day for sanitation reasons. The duration of chemotherapy, on the other hand, was unreported [45]. The EF-11 Phase III trial results did not align with the expectations from the pre-clinical experiments [7]. Although the median survival increased (6.6 months for TTF patients vs. 6.0 months for the control group with the physician’s choice of chemotherapy), the results were not significant enough to prove the effectiveness of TTField therapy (*p* = 0.27) [7,45,49]. Moreover, fourteen patients treated with TTFields experienced radiographic responses, whereas seven control patients experienced similar symptoms; however, the data were not statistically significant [45]. However, 23% of the treated patients did not complete an entire cycle of TTField treatments [49]. The adverse effects reported by patients using TTFileds alone were significantly less than those treated with chemotherapy [1,49].

Numerous post hoc analyses were carried out for the EF-11 phase III trial. For a subgroup analysis of patients under 60 years old, TTFields significantly improved OS [50]. For patients who received at least one complete cycle of TTField treatment, the median OS was significantly higher than that of patients who received at least one complete cycle of control chemotherapy, including bevacizumab (*p* = 0.0093) [50]. Nonetheless, the analysis may have been biased due to the varying durations of one complete cycle of TTFields and chemotherapies, especially with the unknown duration of chemotherapies [45]. Furthermore, patients who complied with at least 18 h of daily TTField treatment (monthly compliance rate ≥ 75%) had a significantly higher median OS compared to those with a compliance rate of less than 75% (*p* = 0.042). The analysis also suggested that the median OS improved with higher compliance (*p* = 0.039) [50]. Patients with prior low-grade glioma, a tumor size ≥ 18 cm^2^, a Karnofsky performance status ≥ 80, and those who had previously failed bevacizumab therapy also displayed a significantly higher median OS with TTField treatments. In modified intention-to-treat (mITT) patients with recurrent GBM, data suggested OS benefits compared to in patients with the physician’s best choice of chemotherapy [50]. Compared to bevacizumab specifically, patients experienced improved survival (6.6 vs. 5.0 months), although the difference was not statistically significant (*p* = 0.054). However, compared to other non-bevacizumab chemotherapies, TTFields did not appear to improve the patients’ OS (6.6 months vs. 6.6 months) [45].

The NovoTTField Patient Registry Dataset is a post-marketing registry dataset comprising 457 recurrent GBM patients in the US who were treated between October 2011 and November 2013 [51]. The average age of the patients was 55, with one-third of the patients being female [45]. Unlike the EF-11 phase III trial, no restrictions were needed for patients’ previous tumor treatments. A total of 55 percent of patients had previously received bevacizumab treatment; 78% had received radiation therapy and chemotherapy; and 63% had undergone debulking surgery. Compared to EF-11 patients, patients treated with TTFields in the patient registry dataset showed an increase in median OS (9.6 vs. 6.6 months), a higher 2-yr survival rate (30% vs. 9%), and a longer duration of treatment (4.1 vs. 2.3 months) and around 10% of patients received TTField treatments for at least two years. Data also suggested that patients with a compliance with treatment of >75% experienced a significantly improved median OS (*p* < 0.0001).

A second prospective trial by Kirson E.D. et al. in 2009 included 20 patients with a histological diagnosis of GBM [45,52]. All patients were older than 18 years old with no previous anti-tumor therapy within the 4 weeks before the TTField treatment. Ten patients were randomly selected to receive TTField treatment alone, while the other 10 received TTField treatments plus maintenance temozolomide (TMZ) after radiation therapy. All patients received treatments for a median duration of 1 year [52]. The median OS of patients treated with TTFields and TMZ was significantly higher than that of patients treated with TTFields alone, suggesting a potential chemotherapeutic efficacy and sensitivity for combination therapy without increasing treatment-related toxicity.

To further analyze the clinical efficiency of TTFields, a second randomized clinical trial, EF-14, was designed to monitor the efficacy of TTFields in conjunction with chemotherapy [53]. EF-14 included 695 patients diagnosed with GBM, including newly diagnosed patients. Of the 695 patients, 229 were treated with radiotherapy plus TMZ, followed by adjuvant TMZ. In comparison, the other 466 patients were treated with TTFields delivered during the adjuvant phase of therapy [45,53]. Thus, the results demonstrated a significant increase in potentiated progression-free survival (6.7 vs. 4.0 months; *p* < 0.001) and median OS (*p* < 0.001), as well as a remarkable improvement in patient survival with the use of TTFields during treatment [53]. The OS for newly diagnosed GBM patients was also significantly better, suggesting that TTFields increased the sensitivity of TMZ [45]. Health-related quality of life did not differ between the treatment groups, except for a notable difference in the incidence of itchy skin. TTFields did not negatively impact patients’ role, emotional, and physical functioning [54]. A subgroup analysis of EF-14 also demonstrated that a minimum monthly compliance rate of 50% of the treatment (TTFields plus TMZ) was necessary to demonstrate improved OS and progression-free survival. A monthly compliance with therapy of 75% or more resulted in a significant increase in the median OS; when the monthly treatment time exceeded 90%, the treatment plan showed its maximum survival benefit (*p* = 0.0007) [55]. However, the EF-14 results showed a promising future for TTFields in combination therapy. Another retrospective study of TTFields analyzed 20 patients treated with TTF and bevacizumab [56]. Compared to historical controls with bevacizumab alone, the median OS of patients treated with TTF and bevacizumab did not show significant differences (*p* = 0.05), indicating the need to test TTFields in combination with other chemotherapies.

## 6. TTFields Dosage and Array Layout

The standard dosage of TTFields is the intensity of TTFields squared, multiplied by the product of tissue-specific conductivities and patient usage [57]. Based on a simulation-based analysis of data from the Phase III EF-14 randomized clinical trial that included 340 enrolled participants, patients showed a significantly longer median progression-free survival (*p* = 0.02) and OS (*p* = 0.003) when the average TTField dosage was ≥0.77 mW/cm^2^ in the tumor bed. Additionally, the data showed a significantly longer median progression-free survival (*p* = 0.02) and OS (*p* = 0.003) when the average TTField intensity was ≥1.06 V/cm, suggesting that an increase in dosage was associated with an improvement in quality of life (*p* = 0.004).

Besides the effects of dosage on TTFields’ effectiveness, various layouts of TTFields also caused different field distributions [58]. TTField array positioning affected median field intensity. For most superficial tumor locations in the cortical regions, the peak of the TTField dose occurred at two separate optimal array positions that were mostly orthogonal. A single optimal array was observed for the maximal field for tumors in deeper subcortical regions. Location-wise, as the tumor’s location moved from posterior to anterior positions, the optimal location of the orthogonal pair of arrays shifted clockwise, all of which agreed with the current placement of the transducer arrays of the Optune device.

## 7. Combination Therapy

### 7.1. TTFields and Chemotherapy

TTFields have been proven to be a promising treatment, especially in combinational therapies. Numerous findings suggest that when combined with TTFields, the effects of chemotherapy are significantly enhanced, possibly due to the increase in BBB and cell membrane permeability that allows the entrance of chemical drugs to reach the cytoplasmic location in cancer cells [1,7,8]. Analyzations of newly diagnosed GBM under MRI screening presented a synergistic effect of TTFields and TMZ chemotherapy, proving that combining the two modalities can more effectively inhibit tumor growth [59,60]. Another combinational treatment method for newly diagnosed GBM adult patients included TTFields, adjuvant TMZ, and pembrolizumab. The study showed a vital survival benefit for the patients undergoing treatments, with a median progression-free survival (PFS) of about 12 months [61]. Though most clinical studies on combinational therapies of TTFields and chemotherapy have only included adult patients, preliminary data have suggested that TTFields are safe and feasible in pediatric patients with high-grade, recurrent glioma and ependymoma [62]. Besides high-grade GBM, combinational therapy with TTFields and chemotherapy has also been shown to be an effective method for other brain tumor treatment. A case report of a 46-year-old male patient with anaplastic astrocytoma, isocitrate dehydrogenase (IDH)1 R132 mutation, IDH2 R172 mutation, and a methylated O6-methylguanine-DNA methyltransferase (MGMT) promoter without 1p36 and 19q13 heterozygosity loss demonstrated the potential of combined TTFields and chemotherapy to reduce the risks for pathological upgrade [63]. Another patient with BRAF V600-mutated high-grade glioma arising from ganglioglioma, after 2 years of combined treatment with dabrafenib and TTFields, revealed “a complete response in all areas with no active lesions or new areas of enhancement” [64].

### 7.2. TTFields and Radiation Therapy

After being exposed to TTFields, cancer cells show increased sensitivity to radiation therapy (RT), with data suggesting that combinational therapy of TTFields and RT leads to more significant DNA damage in cancer cells than radiation therapy alone [65,66,67]. Pre-clinical studies also proposed that TTFields plus radiation therapy trigger synergistic antimitotic effects that eventually lead to cancer cell apoptosis [68]. TTFields also increase the success rate of RT in damaging cancer cells, leading to faster cell death and delayed DNA damage repair [29]. However, it is interesting to note that the additive marks of TTFields on radiation therapy are only effective during the RT treatment, proposing potential in treating RT-resistant cancer cells [29]. A prospective safety and feasibility study was designed to test the combinational therapy of RT, TMZ chemotherapy, and TTFields [69]. Previous studies already suggested that a combinational treatment of RT and TMZ chemotherapy was more beneficial than TMZ alone [60,66]. According to the prospective study, TTFields presented no added toxicity to the combinational RT with TMZ chemotherapy. The clinical results also showed a promising median PFS of 8.9 months with patients outside of the EF-14 study as they presented with early disease progression [69].

### 7.3. TTFields and Targeted Drug Therapy, Immunotherapy, and Others

Sorafenib is a multi-kinase inhibitor used for targeted drug therapy, an anti-proliferative and apoptogenic agent. When combined with TMZ, pre-clinical data showed that the combinational therapy was unsuccessful [8,70]. However, when combined with TTFields, results pointed to decreased cell viability (*p* < 0.05), accelerated apoptosis, and increased autophagy, implying the potential of combination therapy with sorafenib and TTFields in the clinical setting [71]. Another study by Voloshin et al. 2017 also demonstrated the effectiveness of combined therapy of sorafenib and TTFields with a significant reduction in cell count and suppressed tumor growth [72].

As mentioned, TTFileds alone could trigger an anti-tumor immune response in cancer cells. Combined with immune checkpoint inhibitor anti-PD-1, it is shown that this combinational therapy is more effective for treating extra-cranial tumors, implying a potential therapeutic synergy [35,36]. Existing data suggest that combinational therapy with TTFields and anti-PD-1 inhibitor revealed a significant decrease in tumor volume compared to PD-1 inhibitor alone or the control group in vivo [72]. Moreover, other studies indicate that heat therapy and TTFields also have synergistic effects on cancer treatments, with effects including induced apoptosis and more significant inhibition of GBM cell migration and invasiveness [73]. And lastly, combining TTFields with Ca^2+^ antagonists may amplify the effects of each treatment method alone [74].

### 7.4. TTField Effects on Different Cancer Cell Lines

Besides GBM and other brain tumors, TTFields are an efficacious treatment method for different cell lines. Pre-clinical experiments with TTFields effects on pancreatic cancer cells indicated the feasibility of TTField treatments on pancreatic cancers [24]. Further studies have also shown that when combined with chemotherapy, the effects of TTFields were significantly improved for pancreatic cell lines [43]. Other than pancreatic cancers, recent data suggested that TTFields also had effects on lung cancer, brain primary malignancy, prostate cancer, and breast cancer cells [7,41]. In four phase I-II studies in non-small-cell lung cancer (NSCLC), malignant pleural mesothelioma (MPM), pancreatic cancer, and ovarian cancer, data indicated that TTFields at 150–200 kHz showed no severe adverse reactions on treated areas for tested cell lines [75]. For NSCLC, combinational therapy with TTFields and paclitaxel, a chemotherapy agent, revealed a 75% reduction in cell proliferation, suggesting the effectiveness of TTField combinational treatment in lung cancer cell lines [76,77]. Nonresectable MPM is an aggressive cancer type that requires alternating electric fields to be delivered to the thorax [78]. In a phase II clinical trial called STELLAR, combinational therapy with TTFields combined with chemotherapy was clinically tested, with 66% of patients reporting grade 1–2 adverse skin reactions due to TTFields with no other reported severe side effects. The clinical trial concluded that the FDA-approved TTFields combined with chemotherapy are a safe and viable treatment method for nonresectable MPM.

### 7.5. TTField Device

Novocure’s Optune device is currently the primary medical device for treating TTFields in GBM. The FDA approved Novocure for treating recurring GBM in 2011 and expanded the approval to treat newly diagnosed GBM patients in 2015 [7]. The Optune device delivers alternating electric fields to the tumor locations through two pairs of orthogonal transducer arrays [1,44]. Besides the transducer arrays, the package includes a field generator and a power block (a portable battery of 1.2 kg) [8,79,80] (Figure 2). The full package of this device can also be carried around in a compact backpack (Figure 2) and used at home, demonstrating applicability and versatility during the coronavirus pandemic [45]. Due to the quarantine restrictions during COVID-19, TTFields and Novocure’s Optune device displayed the potential to deliver remote care for cancer treatments, especially with the fact that the Optune device enabled the treatments to be carried out outside clinical facilities and monitored remotely [1,81].

This system is a wearable, portable, FDA-approved device for treating GBM in adult patients aged 22 years and older. This figure is reproduced without modification and with permission from [80] under a Creative Commons Attribution 4.0 International License (https://creativecommons.org/licenses/by-nc/4.0/ (accessed on 28 March 2025)).

Since TTFields have no half-life, patients prescribed TTFields are required to wear the device continuously, with wear time exceeding 18 h each day and each cycle lasting approximately 4 weeks to achieve optimal treatment effects [45]. Ideally, the complete treatment plan with TTFields should yield a lower cost than other treatment modalities due to reduced staff time and lower utilization of medical services [45]. Due to the unique nature and locations of each patient’s tumors, establishing a standard care method is nearly impossible for TTFields. An individualized approach is necessary, considering the complexity and numerous unknown factors in the pathophysiology of cranial oncology. In practice, physicians are typically required to generate individualized patient treatment plans, particularly for transducer array positioning, to achieve better reachability and more targeted applications of electric fields [79,82].

### 7.6. Adverse Effects and Contraindications in TTFields or Combination Therapy

Compared to other cancer treatment modalities, TTFields present fewer and less severe side effects and contraindications with no therapeutic resistance [8]. In fact, according to various clinical studies discussed in previous sections, TTFields are often associated with prolonged survival outcomes and efficiently managed and reversible treatment plans [3]. The most significant adverse effect of TTFields is dermatitis, including slight skin irritations and abrasion from the transducer array pads [7] (Table 1). Nonetheless, the skin-related adverse effects can be easily manageable using high-potency topical steroids and cold compress applications [7,45,83]. Since the transducer array pads need to completely fit on the patient’s scalp to deliver the alternative electric fields, the steroids are required to be thoroughly cleaned before the reapplication of array pads, such as using fragrance-free shampoo or 70% isopropyl alcohol to remove the sebum [41,45]. Another way of preventing dermatitis is putting gauze on the areas with a higher likelihood of irritation [7]. Another skin-related side effect is skin infection from bacteria [3]. To manage this side effect, patients must change cranial arrays every 3–4 days. When changing arrays, patients are recommended to wash the scalp carefully, apply topical antibiotics, and shift the position of arrays along with topical treatments on the areas of infection [45,84]. If adverse effects persist, the interruption of treatments for 2–7 days to resolve skin-related symptoms is usually sufficient before resuming the TTField treatment plan [85].

More severe adverse effects include minor scalp burns, minor scalp rashes, and liquefied hydrogel due to high ambient temperatures during TTField delivery, suggesting that TTField devices and apparatus achieve their maximum performance at a cooler temperature [23,45]. High ambient temperature and intense activity may also cause hyperhidrosis, which can be prevented and treated with high-potency topical steroids and the avoidance of using ointments that cause sweating [83]. Less than 1% of the patients under TTField treatments experience skin erosion, severe skin rash, or ulcers, potentially due to irritation or allergic reactions to array pads, sensitivity to shaving hydrogel, moisture, or reaction to tape [45]. The solutions include transducer array removal, wound dressing with gauze, the application of topical antibiotics, and pausing treatments for 2–7 days before device reapplication [83]. TTField treatment must be stopped if the side effects remain severe. Lastly, pruritus, such as dry, itchy, and flaky skin, may also occur during treatments, which can be prevented and treated using non-alcohol-based fragrance-free shampoo products and topical corticosteroids. Less commonly occurring adverse effects include electric shock sensation, array removal during sleep, vivid dreams, confusion with connector ports, and scalp pruritus and meningitis from contemporaneous craniotomy [1,56].

Interestingly, when TTFields were used in combination therapy with adjuvant TMZ after RT plus TMZ, the occurrence of dermatologic adverse reactions became much higher (45%). However, most of them were low-grade [45]. For patients with recurrent GBM, significantly more gastrointestinal, hematologic, and infectious adverse effects were observed with combinational therapy of chemotherapy with TTFields [49,83]. However, for patients with newly diagnosed GBM, no significant increase in systematic adverse effects was observed [83]. Statistically, 45% of GBM patients reported at least one treatment-related adverse effect while receiving TTField treatments. These reactions included 28% skin-related reactions, 9% electric sensations, 4% fatigue, 9% heat sensation, and 5% pain-related reactions [84]. The adverse effects on the nervous system included seizures (8%) and headaches (6%) [84].

## 8. Limitations and Challenges of TTFields Modalities

Compared to other systemic treatment modalities, TTFields propose unique limitations to treatment methods. As discussed previously, TTFields present their maximum functionality when the device is used continuously. Transducer array pads must be applied directly to the skin, and patients must shave their heads every 3–4 days and regularly change the adhesive pads [7,23]. Moreover, the pads must be kept dry, which causes inconvenience during water activities and physical exercise. The advantage is that the device is portable, allowing patients to be mobile and travel. However, the batteries must be charged every few hours to maintain the essential functions of the device [7]. It was reported that many patients were unwilling to initiate the therapy because of the inconvenience of carrying the device continuously to achieve the best optimal effectiveness (TTField treatments require at least 4 weeks (one cycle)) before any tumor inhibition could be observed [41]. Quite often, a more extended treatment period (≥4 weeks continuous treatment) is required to ensure that the treatment reaches the tumor stabilization stage [82]. The long commitment to carrying the device and treatment periods become patients’ most prominent challenge to agreeing to TTField therapy. Also, patients are more likely to experience a psychological burden of appearance changes and reject treatments due to a lack of social support [8,86]. However, analysis suggests that physicians’ initial and follow-up consultations have a profound influence on these parameters and increase patients’ likelihood of agreeing to TTField treatments [87]. Moreover, from a cognitive perspective, there is no evidence to suggest that TTFields will harm patient function and well-being, including role, social, and physical functioning [54].

Another limitation of past TTField treatments was the high cost, which could easily reach USD 21,000 per month in expenses [7]. However, compared to chemotherapy, such as TMZ, alone, TTFields are considered cost-effective for newly diagnosed GBM patients prescribed with combinational therapy or adjuvant therapy with TMZ [60,88]. The current situation has improved with some medical insurance plans covering treatment with TTFields. More challenges are reported for patients with intracranial hardware, such as screws and plates, where the effectiveness of TTFields may be interfered with by the presence of hardware [45]. It is also difficult to distinguish between pseudoprogression and radiation necrosis when analyzing tumor progression after TTField treatments [82].

## 9. Discussion

### 9.1. TTFields and Combination Therapy Comparisons

The therapeutic use of TTFields has shown potential in treating brain oncology, GBM, and other external cancers. This approach offers a non-invasive treatment modality that interferes with tumor cell division. TTField therapy, like any other treatment, despite promising outcomes, presents a spectrum of advantages and challenges that must be considered in clinical practice. In this section, we provide a comparative analysis of TTFields’ various modalities, exploring their advantages and disadvantages to help summarize these treatment methods.

### 9.2. TTField Monotherapy

TTFields have primarily been utilized as a monotherapy for treating GBM, with notable success in improving both PFS and OS in newly diagnosed and recurrent cases [7,45]. The Optune device, a wearable system delivering TTFields, has demonstrated promising results with minimal systemic toxicity, mainly presenting mild dermatologic side effects such as dermatitis, which are reversible with proper management [7,45].

TTField monotherapy has several advantages: (a) Non-invasive—it requires no surgical intervention or systemic drug administration, making it an attractive option for patients seeking a less invasive approach. (b) Offers reduced systemic toxicity—unlike chemotherapy or radiation, which often results in significant systemic toxicity and side effects, TTFields cause localized skin irritations, making it more tolerable for patients [8]. (c) Effective for long-term use—it does not develop therapeutic resistance which is a common challenge in chemotherapy and radiation therapies [8]. Monotherapy with TTFields also has several disadvantages: (a) Adherence to treatment regimen—one of the most significant challenges is the need for continuous treatment (requiring patients to wear the device for more than 18 h per day), which results in patient fatigue, discomfort, and a lack of long-term adherence [45,83]. (b) Limited efficacy for advanced stages—efficacy in treating more advanced or metastatic cancers is still under investigation and more clinical trials are necessary to evaluate its potential in treating other solid tumors [41,76]. (c) Individualized treatment is necessary—due to the variability in tumor locations, TTField treatments require careful customization, including precise array positioning, and therefore, this does increase complexity in clinical implementation [82].

### 9.3. TTFields in Combination with Chemotherapy

TTField therapy in combination with chemotherapy agents (TMZ, paclitaxel, and others) is one of the most promising developments. This combination has been shown to enhance drug delivery to tumor cells, as TTFields increase the permeability of the BBB, improving chemotherapy’s effectiveness [1,8]. Clinical studies, particularly for GBM, have demonstrated that combining TTFields with TMZ leads to significantly improved survival outcomes compared to chemotherapy alone [45]. Thus, a recent study examining TTFields In Germany in Routine clinical care (TIGER) is the largest non-interventional trial for newly diagnosed GBM (ndGBM) patients in Germany. The study aimed to assess the real-world effectiveness and safety of TTFields when added to standard treatments [89,90,91]. As a result of this study, there are a few critical points to note. Survival outcome measures in OS and PFS were significantly improved by adding TTField therapy to adjuvant temozolomide chemotherapy compared to temozolomide alone [90]. Health-related quality-of-life measures did not show a decline in ndGBM patients receiving TTField therapy during the follow-up period, except for reports of itchy skin (citations). Treatment adherence measures demonstrated that a substantial majority (82%) of the 710 surveyed patients chose to continue TTField therapy, demonstrating sustained health-related quality of life, with the primary side effect being increased skin irritation [89].

The combination of TTFields with chemotherapy has some advantages: (a) Enhanced efficacy—the synergistic effect of TTFields and chemotherapy significantly enhances therapeutic efficacy, improving survival rates in GBM and other cancers [7,8]. (b) More targeted delivery—TTFields enhance the delivery of chemotherapeutic agents to the tumor by increasing cell membrane permeability, making them particularly useful for drugs that are otherwise difficult to deliver due to the BBB [1,7]. Disadvantages of this combinational therapy include the following: (a) Increased probability of skin-related side effects—when TTFields are combined with chemotherapy, the incidence of skin reactions, such as dermatitis, increases (45%), and even though these side effects are generally mild, they still pose a challenge to patient adherence and treatment quality [7,45]. (b) Potential for increased systemic toxicity—TTFields’ combination with chemotherapy may lead to more pronounced systemic effects, particularly in the gastrointestinal and hematologic systems [83].

### 9.4. TTFields in Combination with Radiation Therapy

The addition of TTFields to RT has demonstrated the significant enhancement of the effects of radiation, increasing cancer cell sensitivity and promoting apoptosis. TTFields may help delay DNA repair mechanisms in cancer cells, improving therapeutic outcomes in RT-resistant tumors [29,65,66]. This combination has proven beneficial for GBM and other cancers treated with RT.

Advantages of TTField use with RT include the following: (a) Synergistic effects—the combination therapy significantly increases DNA damage and accelerates cancer cell death [65,66], which is particularly effective for RT-resistant tumors [29]. (b) Localized treatment—this combination primarily affects the tumor site, minimizing systemic toxicity and side effects [45]. However, this treatment has a few disadvantages: (a) Acute side effects may lead to increased skin reactions and scalp burns due to the localized delivery of both treatments, although these side effects are usually manageable [45,83]. (b) Limited evidence for long-term efficacy—long-term clinical data are still needed to confirm the durability of this combination therapy, especially for advanced-stage cancers.

### 9.5. TTFields in Combination with Immunotherapy

Recent studies have suggested that TTFields may also play a role in immunotherapy, particularly when combined with immune checkpoint inhibitors (such as anti-PD-1) or chimeric antigen receptor T-cell (CAR-T-cell) therapy. This combination therapy may revolutionize cancer treatment by leveraging the immune system to target and destroy cancer cells with TTFields’ effects. TTFields have been shown to trigger an immune cell response within tumors disrupted by the electric fields, potentially increasing the efficacy of immunotherapy and providing a new avenue for treating tumors unresponsive or intolerant to immuno-based oncologic treatments [72]. This combination shows promise in treating extracranial tumors and other malignancies outside the brain.

Advantages of this treatment include the following: (a) Immune activation—TTFields may enhance the immune system’s ability to target tumor cells by promoting anti-tumor immune responses [72]. (b) Improved efficacy in hard-to-treat cancers—for cancers that are less responsive to chemotherapy or radiation, TTFields combined with immunotherapy offer a novel treatment approach [36]. There are some disadvantages of this combination therapy: (a) Increased risk of side effects—the addition of immunotherapy to TTFields may increase the incidence of side effects, including immune-related reactions such as skin rashes or local inflammation [36,83]. (b) Limited clinical evidence—the combination of TTFields and immunotherapy is still in the early stages of investigation, and more clinical data are needed to better understand the safety and efficacy profile associated with this treatment, especially for long-term treatment plans.

### 9.6. Other Non-Invasive Therapies and TTFields

To assess TTFields’ potential in treating brain oncology, it is critical to compare their applicability and performance and combination therapy, if any, with other available and clinically robust non-invasive modalities. In recent years, treatments such as focused ultrasound (FUS) and proton therapy have gained research and clinical attention as promising alternatives or adjuncts to traditional radiation and chemotherapy.

**Focused Ultrasound.** FUS is a non-invasive treatment modality that uses high-frequency sound waves to target and ablate tumor tissue. Like TTFields, FUS offers a localized treatment option. FUS can also be combined with chemotherapy and immunotherapy, potentially enhancing the delivery of therapeutic agents to the tumor with minimum damage to the healthy tissues outside the target area of treatment [92]. However, FUS requires precise targeting, and its application is generally limited to accessible tumors (such as in the brain or liver) [93]. In contrast, TTFields can be applied to a broader range of cancer types and target deep tumors within healthy tissues. Also, FUS relies more on technology and expertise in focused energy delivery. In contrast, TTFields offer a non-invasive approach that is easier to apply across various tumor types and can be used as a home-based modality.

**Proton Therapy.** It is a form of charged particle therapy that delivers highly targeted radiation to tumors while minimizing damage to surrounding healthy tissue. Proton therapy aims to increase the precision of treatment and reduce side effects. Therefore, it offers superior tumor control in specific tumor types, especially those resistant to traditional therapies or of the base of the skull [94]. Proton therapy has demonstrated clinical efficacy in specific cancer locations; however, balancing high cost, limited availability, and specialized facilities provides substantial challenges in modern healthcare [95]. Proton therapy is also a radiation-based technology, and because of this, it can cause healthy tissue damage over time [96]. In contrast, TTFields offer a non-radiative approach and as such have no radiation-related side effects. Furthermore, they are more accessible and less costly than proton therapy, which makes them a potentially more feasible option in various clinical and home-based applications.

### 9.7. TTFields’ Cost-Effectiveness and Accessibility in Clinical Applications

Cost-effectiveness and accessibility are two significant factors that distinguish TTFields’ therapeutic approach from other cancer therapies. Unlike radiation therapy, immunotherapy, FUS, or proton therapy, which require specialized equipment and high operational costs, TTField therapy is relatively affordable for cancer therapy. TTField devices are portable and can be used in outpatient settings (with the convenience of the home), potentially making them more accessible for patients in different healthcare environments. The NovoTTF-100A device is used in the clinical management of GBM. It is expensive upfront, but given its non-invasive nature and the potential for more prolonged survival and improved quality of life, the overall cost per patient may be competitive compared to other treatments, which makes it a cost-effective alternative. TTField therapy also reduces long-term healthcare costs and related expenses due to the low incidence of severe side effects and low cost to manage adverse reactions, as seen in the EF-11 trial where TTField patients reported fewer and milder adverse events compared to those on chemotherapy [49]. This point also contributes to the cost-effectiveness of TTFields.

Moreover, TTFields’ application does not require the extensive infrastructure setting that proton therapy and immunotherapy require. This improves the accessibility of TTField-related therapeutic approaches. This cost-effective nature makes them a valuable addition to cancer treatment for patients in lower-resource settings where more expensive therapies are not feasible. This also opens a broad spectrum of previously discussed combination therapies to improve treatment outcomes in more clinically challenging tumor types. Due to the wide availability of these treatment modalities everywhere in the nation, FDA approval and coverage by major health insurance plans bring potential regional variations concerning cost-effectiveness to a minimum.

### 9.8. Future Directions of TTField-Related Clinical Applications and Research

Considering recent TTFields’ therapeutic potential, effectiveness, and range of applications, including combination therapies, several avenues for future clinical research and development remain critical.

(a)Optimization of Treatment Regimens

Clinical research may focus on developing a more versatile treatment protocol with optimal treatment duration, frequency, and parameters in TTField therapy to maximize patient outcomes while minimizing side effects. The standardization of treatment protocols could facilitate wider adoption across clinical settings.

(b)Expansion to Other Cancer Types

While TTFields have shown promise in brain cancers, especially GBM, expanding research to other solid tumors (e.g., pancreatic, prostate, breast, and lung cancer) is critical to broadening the applicability of TTFields. Trials exploring combination therapies in these cancers should be prioritized.

(c)Combination with Targeted Therapies

Further exploration into combining TTFields with targeted therapies, like monoclonal antibodies, small-molecule inhibitors, or oncolytic virotherapy, could offer new therapeutic synergies. This research could be particularly important for cancers resistant to conventional treatments.

(d)Personalized Medicine

Because treatments require individualized array positioning and treatment plans based on tumor location and size, future efforts should focus on developing more precise and automated methods for treatment personalization. This would reduce complexity and improve treatment accessibility. More data would be collected, creating a stronger foundation for developing predictive models and artificial intelligence engagement to better predict and understand treatment-related outcomes. This approach would help improve personalized medicine in this field.

(e)Long-Term Clinical Studies

While early studies have shown promising results, long-term clinical trials are necessary to assess the durability of TTField therapy, especially in combination with other modalities such as chemotherapy, radiation, and immunotherapy.

(f)Development of Biomarkers

Clinical research focusing on biomarkers identifying the oncological state of targeted tumors that predict responses to TTFields or combinational therapy could improve patients’ selection and better personalize their treatment. This would allow clinicians to help individualize treatment plans, improving chances for better clinical outcomes and thus enhancing overall treatment efficacy.

## 10. Conclusions

TTField treatments can offer effective non-invasive modalities in the management of brain tumors, including GBM and a variety of other cancers. Their combination with chemo, RT, immunotherapy, or viral oncolytic therapies provides additional tools and options for treating advanced, hard-to-reach, or refractory cancers. This progress comes with challenges regarding adherence to the treatment, side effects, a lack of accessibility in combinational therapy, or the need to develop individualized treatment plans. More work needs to be conducted, and future clinical research focusing on optimized treatment protocols or new combination treatment strategies will be directions for advancing these promising TTFields and their combination alternatives to fully unveil potential in therapeutic modalities in challenging cancer treatment.

## Figures and Tables

**Figure 1 cancers-17-01211-f001:**
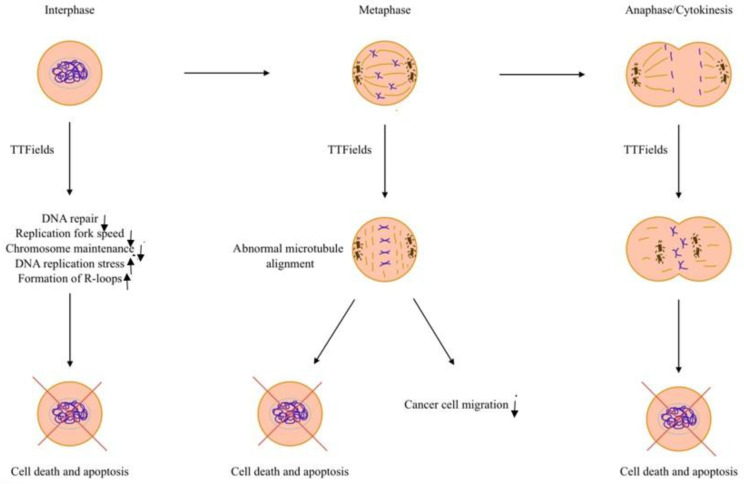
Effects of TTFields on cancer cell biology.

**Figure 2 cancers-17-01211-f002:**
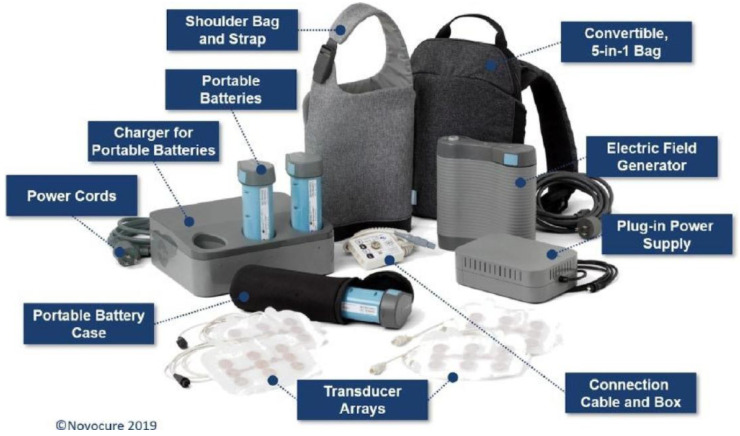
TTField portable therapeutic device—second-generation Optune^®^ system.

**Table 1 cancers-17-01211-t001:** Summary of TTField AEs, potential causes, and suggested interventions.

Adverse Effects	Potential Cause(s)	Treatment Recommendation(s)
Skin irritations/abrasions	Sensitivity to array pads	High-potency topical steroids Gauze to protect areas of irritation Cold, moist compress applications
Skin infection Pustules	Bacterial infection	Change cranial arrays every 3–4 days Wash scalp and shift array positions Topical antibiotics Treatment interruptions for 2–7 days
Hyperhidrosis	Excessive sweating High ambient temperatures Intense activity	Avoid using ointments that cause sweating High-potency topical steroids
Minor scalp burns Minor scalp rashes	High ambient temperatures	Device usage at cooler temperature Liquefied hydrogel
Skin erosion Severe skin rash Ulcers	Allergy to array pads Sensitivity to hydrogel Reaction to tape Mechanical trauma	Treatment interruptions for 2–7 days Transducer array removal Wound dressing with gauze Topical antibiotics
Pruritus Dry, itchy, flaky skin	Cold, dry temperature Dehydration Sensitivity to array pads	Fragrance-free shampoo No alcohol-based products Topical corticosteroids

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
