# Peer review of "Tumor Treating Fields and Combination Therapy in Management of Brain Oncology"

_cancers, 2025, doi:10.3390/cancers17071211_

Round 1

Reviewer 1 Report

Comments and Suggestions for Authors

The work is a comprehensive review of TTFields treatment . Perhaps a little too long.
Given the purpose of the review, I think it would be appropriate to include a picture of the device.
In the economic sustainability and cost benefit part of the treatment, cost effective should be added which, albeit with regional variations, may direct the reader.
The section on prototherapy and focused ultrasound seems superfluous in this context. 

I think it would be useful to expand on the findings of the German Tiger study

Author Response

Comments and Suggestions for Authors:

The work is a comprehensive review of TTFields treatment . Perhaps a little too long.
Given the purpose of the review, I think it would be appropriate to include a picture of the device.
In the economic sustainability and cost benefit part of the treatment, cost effective should be added which, albeit with regional variations, may direct the reader.
The section on prototherapy and focused ultrasound seems superfluous in this context. 

I think it would be useful to expand on the findings of the German Tiger study.

Response:

Thank you for your positive and comprehensive feedback on improving this review paper, with a focus on TTFields treatment and making it reader-friendly and complete.

Comment: -Given the purpose of the review, I think it would be appropriate to include a picture of the device.

Response: The authors agree with the reviewer’s valid point, and a picture of the TTFields device has been added to the manuscript (please see Figure 2 with its figure legend). The authors considered adding different device representations from a non-open-access source. However, it took the copyright owner too long to respond and grant permission to publish the copyright.

Comment: -In the economic sustainability and cost benefit part of the treatment, cost effective should be added, which, albeit with regional variations, may direct the reader.

Response: The authors agree with the reviewer, and Section 9.7 of the manuscript has been revised accordingly to address the requested information.

Comment: -Perhaps a little too long. The section on prototherapy and focused ultrasound seems superfluous in this context.

Response: The authors carefully evaluated the comment and respectfully believe that some readers may benefit from having Focused Ultrasound and Proton Therapy topics discussed in the context of TTFields, to provide a more comprehensive overview of other non-invasive modalities in the management of brain oncology.

Comment: -I think it would be useful to expand on the findings of the German Tiger study.

Response: The authors agree that the German TIGER study could benefit readers in the context of this review paper and have therefore added essential highlights of the study. Please see Section 9.3 for the newly added text as requested.

Reviewer 2 Report

Comments and Suggestions for Authors

The manuscript is interesting and well organized. However major revision is required for publication. 

Comments:

  1. The authors should highlight the new and alternative therapies in the research field of glioblastoma in the Introduction section focusing on recent preclinical studies.
  2. The authors should make a table of the main articles cited in the manuscript.
  3. The authors should highlight the future perspectives 
  4. The authors should revise the text to avoid typos. 

Author Response

Comments and Suggestions for Authors:

The manuscript is interesting and well organized. However major revision is required for publication.

Comments:

  1. The authors should highlight the new and alternative therapies in the research field of glioblastoma in the Introduction section focusing on recent preclinical studies.
  2. The authors should make a table of the main articles cited in the manuscript.
  3. The authors should highlight the future perspectives 
  4. The authors should revise the text to avoid typos. 

Comment: -The manuscript is interesting and well organized. However major revision is required for publication.

Response: Thank you for your time, expertise, and detailed approach to improving this comprehensive review. The authors greatly appreciate your positive, constructive, and supportive feedback!

Comment:  -The authors should highlight the new and alternative therapies in the research field of glioblastoma in the Introduction section focusing on recent preclinical studies.

Response: Thank you for this comment! The authors agree that highlighting the new and alternative modalities in managing GBM from recent preclinical studies would significantly benefit readers. A new section was added to the Introduction to address this constructive comment.

Comment: -The authors should make a table of the main articles cited in the manuscript.

Response: Thank you for the comment! Due to the size of the review and page limit, as well as the main cited articles in the reference list, the authors agreed that the primary references list will comprehensively serve the purpose well. Authors leave this to the journal editor's discretion on how they would like to manage references.

Comment: -The authors should highlight the future perspectives.

Response: The authors agree with the valid point and appreciate constructive feedback. Section 9.8. of this manuscript is dedicated to future directions in clinical applications and research linked to TTFields. This section should address the reviewer’s comment.

Comment: -The authors should revise the text to avoid typos.

Response: The authors are thankful for the feedback and have revised the text to address typos and limitations in the manuscript's abbreviations.

Reviewer 3 Report

Comments and Suggestions for Authors

Nice review but lacks Figures and the Figure shown lacks a Figure legend

Author Response

Comments and Suggestions for Authors

Nice review but lacks Figures and the Figure shown lacks a Figure legend

Response: The authors are thankful for the reviewer’s time and feedback in improving this work. The authors agreed that adding another figure and completing the figure legends would complement the content and greatly benefit readers. Therefore, Figure 2, along with its corresponding Figure Legends, was included in the main text.

The authors considered adding different device presentations from a non-open-access source. However, it took the copyright owner too long to respond and grant permission to publish the copyright.

Round 2

Reviewer 2 Report

Comments and Suggestions for Authors

The authors improved the manuscript